# Modeling Key Characteristics of Rigid Polyisocyanurate Foams to Improve Sandwich Panel Production Process

**DOI:** 10.3390/ma18040881

**Published:** 2025-02-17

**Authors:** Mikelis Kirpluks, Beatrise Sture-Skela, Uldis Bariss, Iveta Audzevica, Uldis Pasters, Nikolajs Kurma, Laima Vēvere

**Affiliations:** 1TENAX PANEL Ltd., Str. Spodribas 1, LV 3701 Dobele, Latvia; uldis.bariss@tenaxgrupa.lv (U.B.); iveta.audzevica@tenaxgrupa.lv (I.A.); uldis.pasters@tenaxgrupa.lv (U.P.); nikolajs.kurma@tenaxgrupa.lv (N.K.); 2Polymer Laboratory, Latvian State Institute of Wood Chemistry, Str. Dzerbenes 27, LV 1006 Riga, Latvia; beatrise.sture@kki.lv (B.S.-S.); laima.vevere@kki.lv (L.V.)

**Keywords:** rigid polyisocyanurate, response surface modeling, foaming kinetic parameters

## Abstract

This study explores the optimization of rigid polyisocyanurate (PIR) foam formulations, focusing on foaming kinetics that significantly influence the foam’s microstructure and thermal insulation properties. By systematically altering components such as isocyanate, polyols, catalysts, blowing agents, and additives, this research investigates their effects on key characteristics including cell density, mechanical strength, and thermal conductivity. A statistical approach known as response surface modeling (RSM) was employed to identify relationships between formulation variables and performance metrics. The optimization aimed to enhance thermal insulation while ensuring feasibility for industrial-scale production, particularly for sandwich-type PIR panels. Two distinct formulations, with isocyanate indices of 335 and 400, were developed to assess the impact of various parameters on properties like foaming start time, gel time, and density. The results indicated that the choice of blowing agents and catalysts played a pivotal role in controlling foaming kinetics and final mechanical properties. The optimized formulations exhibited competitive thermal conductivity values (around 23.7 mW/(m·K)) and adequate compression strength (0.32 MPa), aligning closely with commercially available materials. These findings affirm the potential for enhancing production efficiency and performance consistency in the manufacturing of rigid PIR foams for insulation applications.

## 1. Introduction

Thermal insulation for buildings refers to the use of materials and construction techniques designed to reduce the transfer of heat between interior and exterior environments. The primary purpose of thermal insulation is to maintain a comfortable indoor temperature, enhance energy efficiency, and reduce heating and cooling costs.

The benefits of thermal insulation include energy efficiency. An effective insulation envelope reduces the amount of energy required to heat or cool a building, leading to lower energy consumption. A well-insulated building maintains a more consistent indoor temperature, improving comfort for residents. Reduced energy consumption leads to lower greenhouse gas emissions, contributing to environmental sustainability. Many insulation materials also have soundproofing qualities, reducing noise pollution from outside, such as aerogels [1,2,3], textiles [4], and foam-based insulation [5]. Lastly, proper insulation can help prevent moisture buildup [6,7,8], reducing the risk of mold and structural damage.

There are several types of thermal insulation materials that are commonly used on the market. Most common insulation materials can be split into mineral fiber insulation, foamed polymer insulation, and biomass-based insulation. Each has its pros and cons [9].

Mineral wool made from basalt or slag is one of the most common insulation materials. It can withstand high temperatures and is used for thermal and sound insulation [9]. In a similar application, a fiberglass made of fine glass fibers is applied. Both types of insulation are effective, relatively inexpensive, and non-combustible. However, they lack water resistance [10], their application is labor-intensive, and the bulk material production is energy-intensive [11].

Foam insulation includes materials like polyurethane (PU) foam, expanded polystyrene foam (EPS), extruded polystyrene foam (XPS), and polyisocyanurate (PIR) foam. These are available as rigid boards, spray foam, and insulating concrete forms. The main benefit of polymer foam insulation is its superior thermal insulation performance and relatively low price [12]. Polymer foam’s main drawback is its flammability, which is mitigated by the liberal application of various flame retardants, such as tris(chloropropyl) phosphate (TCPP) [13], tris(ethyl) phosphate (TEP) [14], and brominated flame retardants [15]. Furthermore, the majority of polymer foam insulation is produced using petrochemical feedstocks, which increases their environmental footprint. Although there are plenty of studies describing rigid PIR foam development from bio-based feedstocks such as rapeseed oil [16], tall oil fatty acids [17], and date seed oil [18], their wider application on an industrial scale is limited due to their cost. Moreover, polymer foams have higher water vapor resistance than fiber-based insulations [12] (both mineral and bio-based), which could lead to moisture retention inside the building if ventilation solutions are not implemented.

Lastly, the various bio-based thermal insulation materials allow for a potential reduction in the CO_2_ footprint associated with the production of thermal insulation. Cellulose fibers are made from recycled paper products and are heavily treated with fire retardants [19]. Other natural insulations include materials like sheep’s wool [20,21,22], cotton fiber [23], wheat straw [24], and hemp fibers [25]. These are sustainable but may be more expensive.

A comparison of the main thermal insulation characteristics, i.e., thermal conductivity and its apparent density in the different thermal insulation materials, is summarized in Table 1. All modern thermal insulation materials have a thermal conductivity below 81.5 mW/(m·K) [23]. Among these materials, foamed polymer-based insulations stand out as they are nearly twice as effective as fiber-based insulation materials. This superior performance can be attributed to their distinct structural characteristics. Foamed polymers typically possess a low apparent density and a closed-cell structure, which minimize air movement within the material and significantly reduce thermal conductivity. In contrast, fiber-based insulation materials, while still effective, rely on open or semi-open structures that allow more air movement, limiting their insulating efficiency compared to foamed polymers.

This study focuses on the development of rigid PIR foam that can be applied as the core of the sandwich panel structure. Rigid PIR foam is usually used as the sandwich panel insulation between two metal sheets. Rigid PIR foam sandwich panel insulation offers numerous benefits, making it a popular choice in various construction and industrial applications. Excellent thermal insulation provides energy efficiency, reducing the need for artificial heating and cooling. This leads to significant energy savings and lower utility bills. Despite being lightweight, rigid PIR foam sandwich panels offer a high strength-to-weight ratio, making them ideal for applications where weight reduction is crucial without compromising structural stability, such as warehouse wall construction or roofs with large surface areas. Furthermore, their lightweight nature makes them easy to transport, handle, and install, reducing labor costs and construction time. PIR sandwich panels come in pre-fabricated sections, which can be quickly assembled on-site. This speeds up the construction process significantly, and the ease of installation reduces the need for skilled labor and shortens project timelines, leading to cost savings. PIR sandwich panels are made from durable materials and can withstand harsh weather conditions, impacts, and other environmental factors. They require minimal maintenance, which translates into lower long-term upkeep costs. Lastly, PIR sandwich panels are designed with fire-resistant cores, enhancing the safety of buildings by slowing the spread of fire. They can meet stringent fire safety regulations and standards, making them suitable for use in areas where fire safety is a critical concern compared to PU and EPS foams.

The objective of this study was to investigate the influence of various rigid PIR foam formulation parameters on the common characteristics of foamed insulation material. The foam rise kinetics are crucial for obtaining high-quality material. Rigid PIR foam sandwich panels are produced by pouring a reacting PIR mass between two metal sheets. Furthermore, the apparent density of the material is a key factor determining the economic feasibility of the produced material. Rigid PIR foam is created by mixing several components, such as a polyol mixture, blowing catalyst, trimerization catalyst, and blowing agent. This study examined the influence of these parameters on the properties of rigid PIR foams, including foaming kinetic parameters, apparent density, thermal conductivity, and compression strength, using the response surface modeling (RSM) approach. Rigid PIR foams with two different isocyanate indices of 335 and 400 were developed. RSM facilitated the selection of optimal rigid PIR foam formulations for further upscale and material production in a sandwich panel production line.

## 2. Materials and Methods

The following reagents were used to develop rigid PIR foam. Low-functionality polyol with an OH value of 227.1 mg KOH/g, water content 0.59%, and high-functionality polyol with an OH value of 312.9 mg KOH/g, water content 0.03%, were supplied by TENAX PANEL Ltd. (Dobele, Latvia) and used after OH value and water content determination according to DIN 53240-2 and DIN 51777 standards, respectively. Flame retardant, triethyl phosphate (TEP), a silicone surfactant with trade name Additive 9968, amine blowing catalyst, potassium-based trimerization catalyst, n-pentane physical blowing agent, and polymeric methyl-diphenyl diisocyanate (pMDI) were supplied by TENAX PANEL Ltd. (Dobele, Latvia) and used as received. Lastly, deionized water was used as a chemical blowing agent.

The preparation of PIR foams and the characterization of foaming parameters were performed using specialized equipment. PIR foam was obtained in a one-step method by mixing the isocyanate component (pMDI) with a mixture of the polyols, catalyst, surfactant, water, and physical blowing agent. After mixing for 5 s with a mechanical stirrer, the reaction mixture was quickly poured into a FOAMAT^®^ (Karlsruhe, Germany) advanced test container (ATC), allowing the free foam to rise in the horizontal foam direction. The temperature in the ATC was 60 °C.

The parameters of the foaming process, such as the dielectric polarization of the reaction mixture, the foam core temperature, the foam’s height, and the foam rise pressure, were analyzed using the FOAMAT^®^ device. These parameters illustrate the reactivity of the PIR system.

Larger samples were prepared in an open-top mold using the same formulations and mixing time as for FOAMAT^®^ testing. The foams were conditioned for two hours at 60 °C and 22 h at room temperature.

For the characterization of PIR foams, the apparent density of the obtained PIR foams was measured according to ISO 845:1995 [35].

The thermal conductivity coefficient of PIR foams was analyzed using an A FOX 200 (TA instruments-Water LLC; Lukens Drive, New Castle, DE 19720, USA) according to ISO 8301:1991 [36], using samples with dimensions of 200 mm × 200 mm × 50 mm after 24 h of foam preparation. During the thermal conductivity measurement, a one-way heat flow between the hot (20 °C) and cold (0 °C) plates was established at an average temperature of 10 °C between the two plates.

Compressive strength tests were carried out using a Zwick/Roell Z010 (10 kN) static material testing device (Zwick Roell, Ulm, Germany) coupled with a 1 kN force cell according to ISO 844:2014 [37] in directions parallel and perpendicular to the direction of foam rise. Instead of standard samples, cylindrical samples with a 20 mm diameter and a 22 mm height were used.

Using Design Expert software (Version V12.0.7.0, Stat-Ease, Inc., Minneapolis, MN, USA), an experimental matrix was developed to analyze the effects of four factors on the properties of rigid PIR foam.

## 3. Results and Discussion

The characteristics of rigid PIR foams are governed by a wide range of factors, with foaming kinetics being one of the most critical. Foaming kinetics refers to the rate at which the foam expands and solidifies during production, which affects the foam’s microstructure and thermal insulation properties. The rate of cell formation, growth, and stabilization determines key features such as cell size, cell density, and the uniformity of the foam, all of which are closely tied to the material’s thermal performance.

Changes in the PIR foam formulation, including the ratios of isocyanates, polyols, catalysts, blowing agents, and additives, can significantly alter the foaming kinetics and the resulting polymer matrix. These adjustments impact not only the foam’s structural integrity and insulation capacity but also its mechanical properties, such as compressive strength and thermal conductivity. For instance, varying the blowing agent concentration may change the foam’s density, leading to improved or diminished thermal conductivity.

In this study, the formulation of rigid PIR foam was systematically modified by altering specific components to better understand their effects on the material’s behavior. The objective was to identify an optimal formulation that enhances the foam’s thermal insulation characteristics while ensuring scalability for industrial applications, particularly in the production of sandwich-type rigid PIR panels, which are widely used in construction due to their lightweight and high insulating efficiency.

To achieve the objectives, RSM was employed as a statistical approach to explore the relationships between the formulation variables and the foam’s key performance metrics. By using linear and polynomial models, RSM allowed for the identification of both linear and non-linear interactions between variables. These models provided insight into how individual components—such as the type and concentration of polyols, catalysts, and blowing agents—influence the final properties of the foam, including foaming kinetic parameters, apparent density, thermal conductivity, and compressive strength. The optimization process aimed to fine-tune the balance of these parameters to maximize performance, with the added goal of ensuring the formulation could be easily adapted to industrial-scale production processes.

This in-depth analysis not only provides a pathway to improved PIR foam formulations but also demonstrates the potential for scaling these optimizations to the large-scale manufacturing of sandwich panels, where consistent quality and performance are critical.

### 3.1. Rigid PIR Foam Formulation

The formulation of rigid PIR foam includes several key components: polyols, flame retardants, surfactants, water (as a chemical blowing agent), gelling catalysts, trimerization catalysts, physical blowing agents, and the isocyanate component pMDI. During the production of sandwich-type panels, these components are mixed in the production line, with different methods available for combining them. The components can either be mixed dynamically, statically in-line, or premixed before final mixing in the production line. The isocyanate component is typically added last, usually in the mixing head.

The mixing process in the production line can occur under varying pressure conditions, depending on the design of the specific production setup. Some lines incorporate high and low pressure zones to accommodate the different stages of component mixing. Additionally, the formulation of rigid PIR foam is adjusted based on the thickness of the panels being produced. This adjustment often involves altering the catalysts and the amount of blowing agent used. Therefore, it is crucial to evaluate how different component compositions affect key kinetic parameters of the foaming process, such as foaming start time, gel time, tack-free time, and rise time. These different components also impact the physical properties of the rigid PIR foam, including apparent density, thermal conductivity, and compressive strength.

In this study, two different rigid PIR foam formulations were developed, each with a different isocyanate index (*II*): 335 and 400. The isocyanate index is calculated as the ratio of OH groups to NCO groups multiplied by 100. An isocyanate index greater than 100 indicates an excess of NCO groups, allowing for the formation of isocyanate trimerization products. The resulting PIR rings are more thermally stable than urethane groups, and they provide higher mechanical properties due to a more crosslinked polymer matrix.

However, a higher isocyanate index also presents challenges. The isocyanate component is more expensive, and formulations with a high isocyanate index are more difficult to optimize during production due to stringent temperature control requirements. Rigid PIR foams with an isocyanate index above 400 are considered polyisocyanurate foams, as they contain both urethane and polyisocyanurate groups. The formation of polyisocyanurate rings requires three isocyanate groups, a specific catalyst, and an internal foam temperature exceeding 140 °C.

The developed rigid PIR foam formulations are depicted in Table 2. The four changed parameters were the low functionality polyol content, the two different catalyst content, and the physical blowing agent content in the rigid PIR foam formulation. The parameter influence on the rigid PIR foam foaming kinetics and foam characteristics was evaluated by RSM.

### 3.2. Modeling of the Rigid PIR Foam Characteristics Using RSM

The data were analyzed using multiple regression techniques to develop an RSM. The influence of altered factors on the selected responses was approximated using either linear or second-order polynomial models, with a 95% confidence level. The models were further refined by removing non-significant terms (*p*-value greater than 0.05). Due to the mutual dependence between low-functionality polyol (LF-polyol) and high-functionality polyol content in the rigid PIR foam formulations, only one polyol content could be used as a factor in the model development within the experimental matrix. The changed parameters, their identifier, their lower and higher values, and coded levels are summarized in Table 3.

The models developed for rigid PIR foams and their analysis of variance (ANOVA) with an isocyanate index of 335 are shown in Table 4, while the models for PIR foams with an isocyanate index of 400 are shown in Table 5. In total, 18 different models were created to characterize the relationship between rigid PIR foam properties and the varied parameters. A similar approach was performed by H. Li et al. [38] in the development of bio-based PU foams from liquefied wheat straw. The measured data of the different responses and the changed parameters are summarized in Appendix A.

The statistical strength of the models was confirmed by their F-values, which were consistently above 10.0, indicating that the models are highly significant. In simple terms, a high F-value indicates that the variation explained by the model is much greater than the unexplained variance (noise). The probability of obtaining such high F-values by chance was extremely low—only 0.01%—which further solidifies the robustness of the models. The only exception was thermal conductivity, where the F-value ranged between 5 and 10. Although slightly lower, these values still indicate statistical significance, as all model *p*-values remained well below the 0.05 threshold.

Additionally, the R^2^ values of the models—which measure the proportion of variance in the responses explained by the model—were strong, above 0.9 for rigid PIR foams with an isocyanate index of 335. High R^2^ values suggest that the models can account for the majority of data variation, making them reliable tools for navigating the design space and optimizing key properties of the rigid PIR foams. Essentially, these models provide a clear path to understanding how different formulation variables impact the final foam characteristics, enabling targeted optimization for specific performance goals.

In the subsequent sections, the developed models are depicted as topographic planes, offering a visual representation of how the response variables (such as foam density or thermal conductivity) change with variations in the formulation factors. The experimental data points are also overlaid on these models, represented by red or pink dots, to illustrate the correspondence between the model predictions and the actual observed results. This visual approach further emphasizes the models’ accuracy and reliability in capturing the experimental data’s underlying relationships.

### 3.3. Rigid PIR Foam Foaming Kinetic Parameters

The foaming kinetic parameters were measured using FOAMAT^®^ equipment, and the RSM results for the foaming start time of rigid PIR foams with isocyanate indices of 335 and 400 are shown in Figure 1. This study revealed that the physical blowing agent (n-pentane) and the LF polyol content had minimal influence on the foaming start time of the rigid PIR foam (see Figure 1a,b). In contrast, the contents of the trimerization and blowing catalysts had a much more significant impact (see Figure 1c,d).

Notably, the blowing catalyst reduced the foaming start time approximately 1.5 times more effectively than the trimerization catalyst. The regression coefficients presented in Table 4 and Table 5 quantify the influence of both catalysts on the foaming start time. Additionally, the blowing and trimerization catalysts demonstrated a synergistic effect, with their combined presence further influencing the PIR foam foaming start time.

The limited effect of n-pentane and LF polyol on the foaming start time is particularly important, as it simplifies the scalability of the developed formulations. This allows the foaming start time to be controlled primarily through adjustments in the catalyst concentrations, streamlining the manufacturing process for rigid PIR foams.

In contrast to the foaming start time, the gel time of rigid PIR foam was significantly influenced by the LF polyol content. Figure 2a,b show that reducing the LF polyol content resulted in a shorter gel time. This effect is attributed to the use of higher-functionality polyols, which have a higher OH value (312.9 mg KOH/g) compared to LF polyols (227.1 mg KOH/g). The higher-functionality polyol increased the crosslinking density of the PIR foam polymer matrix, accelerating the gelling process. Gel time is a critical parameter in the production of rigid PIR foam, as the material must gel just before reaching the double-belt press. If the PIR foam gels too early, it will not adhere properly to the metal sheet. Conversely, if it has not gelled by the time it enters the press, the foam may be crushed, compromising its structural integrity and material properties.

The physical blowing agent content had only a minimal effect on gel time. As anticipated, the blowing and trimerization catalysts had the most significant impact, as shown in Figure 2c,d. The trimerization catalyst was approximately 2.5 times more effective in reducing gel time than the blowing catalyst.

To ensure optimal production, gel time on the PIR foam production line should be carefully controlled by adjusting the concentrations of blowing and trimerization catalysts, depending on the sandwich panel thickness and production line speed. Additionally, the choice and combination of polyols play a crucial role in achieving the desired gel time, ensuring efficient processing and high-quality end products.

The remaining two foaming kinetic parameters for rigid PIR foam—rise time and tack-free time—are summarized in Appendix A, respectively. The physical blowing agent, LF polyol content, and the two catalysts influenced these parameters in a manner similar to their effects on the gel time of rigid PIR foam.

Additionally, the temperature peak observed in the PIR foam core during the foaming process is shown in Appendix A. The physical blowing agent, which absorbs heat during evaporation, contributed to a reduction in the temperature peak. The LF polyol, with a lower OH group content available for reaction with isocyanate, generated less heat during the reaction. The two catalysts had only a marginal effect on the temperature peak.

Despite these variations, the temperature peak ranged between 170–185 °C, which is optimal for rigid PIR foams. Maintaining the temperature within this range is critical; temperatures exceeding 200 °C risk causing thermal degradation of the foam, while temperatures below 140 °C are insufficient for the formation of isocyanurate rings, which are essential for the foam’s structural integrity and performance [39].

### 3.4. Rigid PIR Foam Thermal Conductivity Properties

The initial thermal conductivity of rigid PIR foam, measured 24 h after preparation, is shown in Figure 3. Rigid PIR foams with an isocyanate index of 335 exhibited slightly lower thermal conductivity compared to those with an isocyanate index of 400 (see Figure 3a,b). This difference can be attributed to the slightly lower apparent density of the foam with an isocyanate index of 335, as shown in Figure 4. Although the blowing agent content remained constant, the higher isocyanate content in the foam with an index of 400 resulted in more of the polymer matrix in the PIR foam volume, which increased thermal conductivity.

Heat transfer in rigid PIR foams is governed by three main mechanisms: heat transfer through the gas inside the cells, heat transfer through the polymer matrix, and radiative heat transfer. In these foams, the polymer matrix typically accounts for only 15–25% of the total heat transfer, while the majority (75–85%) occurs through the gas or blowing agent trapped within the closed-cell structure [40,41]. The polymer’s contribution to heat transfer is relatively low because, although it has higher thermal conductivity than the gas, it occupies a much smaller volume fraction in the foam. Optimizing the cell structure and selecting low-conductivity blowing agents are therefore crucial for enhancing the thermal insulation properties of rigid PIR foams. Radiative heat transfer is generally negligible in this context.

The type and content of catalysts used in the preparation of the PIR foams had little impact on their thermal conductivity. Notably, the thermal conductivity of the developed PIR foams ranged from 23.0 to 23.8 mW/(m·K), a very small variation. These values indicate that the foams possess excellent thermal insulation properties, making them suitable for use as insulation materials.

When produced on a sandwich panel production line, the morphology of the rigid PIR foam is expected to change due to improved mixing of the polyol and isocyanate components in a high-pressure mixing head. This improved mixing will result in smaller cell sizes. Reducing the cell size in PIR foam can significantly improve its thermal insulation properties by decreasing its thermal conductivity. As the cell size decreases, the foam’s structure becomes more effective at trapping gas, which has a lower thermal conductivity compared to the polymer matrix itself. This reduction in cell size limits the amount of heat transfer through the gas phase, thus improving the foam’s overall thermal resistance [42,43].

Typically, reducing the cell size in rigid PIR foams can lead to a decrease in thermal conductivity by approximately 10–20%. The exact reduction depends on various factors such as the foam density, the type of blowing agent used, and the processing conditions by as much as up to 4 mW/(m·K) [44,45]. Bringing the values of developed PIR foams closer to 19–20 mW/(m·K) compared to the initial range of 23–24 mW/(m·K). Therefore, controlling and reducing the cell size is a key strategy for enhancing the thermal efficiency of rigid PIR foams and will be carried out during the upscaling of material production.

The changed parameters significantly impacted the apparent density of rigid PIR foams, as shown in Figure 4. As expected, increasing the physical blowing agent (n-pentane) reduced the apparent density of the rigid PIR foam for both isocyanate indices (335 and 400). Similarly, an increase in the LF polyol content also reduced the apparent density, as it lowered the isocyanate amount in the foam formulation due to the polyol’s lower OH value (see Figure 4a,b).

For rigid PIR foams with an isocyanate index of 335, the two different catalysts did not affect the material’s apparent density (see Figure 4c). An unexpected result was observed in the effect of the blowing and trimerization catalysts on the apparent density of rigid PIR foams. The “pigs’ ear” effect, shown in Figure 4d, indicates that at low concentrations of these catalysts, the foam’s reactivity is also low (see Figure 1d and Figure 2d), resulting in insufficient heat generation for foam expansion (see Appendix A). However, when the blowing catalyst content reached 4 pbw, and the trimerization catalyst content reached 2.5 pbw, the reactivity of the PIR foam with an isocyanate index of 400 became excessively high. This caused the foam to gel before it could fully expand, resulting in a second peak in apparent density within the experimental range.

The apparent density of the foam must be carefully optimized, as it directly impacts the cost-effectiveness of the thermal insulation material. A higher apparent density increases raw material consumption, which raises production costs. Additionally, it results in higher shipping and delivery costs for finished sandwich-type panels.

### 3.5. Rigid PIR Foam Compression Properties

The influence of the modified parameters on the compression strength and compression modulus of the rigid PIR foams is presented in Figure 5 and Figure 6, respectively. The physical blowing agent, n-pentane, reduced the compression properties of the PIR foams by decreasing the apparent density of the material. This is because the compression strength of rigid foams is closely linked to their density; as the density decreases, the material becomes less able to resist compressive forces. Similarly, the LF polyol reduced the compression strength of PIR foams with both isocyanate indices (335 and 400) due to its lower OH value, which reduced the overall isocyanate content in the formulation. This led to a lower crosslink density and, consequently, a reduction in the foam’s structural rigidity and apparent density, as shown in Figure 4.

Interestingly, the catalyst content had only a marginal effect on the compression properties of the developed rigid PIR foams. The tested catalyst levels were sufficient for the PIR foam formulations to fully cure, and the chemical composition of the polymer matrix remained consistent across samples. This indicates that the compression strength and modulus were more dependent on density-related factors than catalyst variations. The “pigs’ ear” effect seen in Figure 4d is not that prevalent for the compression strength properties. However, more data are necessary to fully display the apparent density influence on the compression properties of the developed PIR foams.

Compression strength is a critical characteristic for the intended application of PIR foams as core materials in sandwich-type panels. These panels serve as load-bearing structures, where the foam core provides mechanical support and thermal insulation, while the metal sheets contribute to the overall stiffness and structural integrity. For the panels to perform effectively, the compression strength of the PIR foam core must meet or exceed 0.08 MPa to withstand applied loads without deformation or failure [46]. This benchmark was achieved across all tested formulations, demonstrating that the developed PIR foams are suitable for use in thermal insulation applications.

Furthermore, achieving the target compression strength is essential for optimizing the material’s performance-to-cost ratio. Reducing the density of the foam while maintaining adequate compression strength allows for lower raw material usage, reduced production costs, and decreased transportation expenses. These benefits make the developed PIR foams not only effective but also economically viable for industrial-scale production.

### 3.6. Optimization of Rigid PIR Foam Formulations

Desirability in RSM is a statistical technique used to simultaneously optimize multiple responses or objectives. It works by converting each response into a dimensionless scale called a “desirability function”, which ranges from 0, indicating a completely undesirable outcome, to 1, representing a completely desirable outcome. The goal is to maximize the overall desirability, which reflects how well the responses collectively meet the set criteria.

Each response is transformed into an individual desirability value (*dᵢ*) using a specific desirability function. If a response value meets the target, its desirability is equal to 1, while responses falling outside acceptable limits are assigned a desirability value of 0. Between these extremes, the desirability increases or decreases based on the proximity of the response to the desired target. The shape of the desirability function depends on the optimization goal for a specific response. For instance, when the objective is to maximize, the desirability increases as the response approaches a higher value. Conversely, desirability increases when minimizing as the response approaches a lower value. The desirability is highest at a precise target value for target-specific objectives and diminishes as the response deviates from it.

The optimization goals, their ranges, and the importance parameter of each factor are detailed in Table 6. A different level of importance was assigned to each factor or response to prioritize the more critical ones. The importance was scaled on a range from 1 to 5, with 1 indicating the least significance and 5 representing the highest significance. The overall function is reduced to 0 if any factor or response falls outside its desirable range. One of the key desired parameters was the trimerization catalyst content, set at 2 pbw, as it plays an essential role in curing rigid PIR foam. The foaming kinetic parameters of the PIR foam were optimized to meet industrial production criteria for PIR sandwich panels, as specified by TENAX PANEL Ltd. The start time, gel time, and rise time were set at 10.9 s, 37 s, and 57 s, respectively. These values are consistent with those used in the current production line and ensure compatibility with industrial-scale manufacturing.

In addition to the kinetic parameters, the apparent density of the foam was minimized to develop a cost-effective material. At the same time, thermal conductivity and compression strength were maximized to create a rigid PIR foam with competitive thermal insulation and mechanical properties. This optimization approach ensures the resulting material meets industrial standards while remaining economically viable.

Overall desirability (*D*) is when multiple responses are involved, and the individual desirability values, *dᵢ,* are combined into an overall desirability index, *D*, using a geometric mean. *D* ranges from 0 to 1, where *D* = 1 indicates that all responses have achieved their most desirable values. The *D* of the developed PIR foams is depicted in Figure 7, and it shows a relatively narrow area of the different parameters that meet all of the required criteria. The maximum *D* was selected for the most optimal rigid PIR foam formulation for foams with isocyanate indices 335 and 400, and the material was produced and tested. The optimized rigid PIR foam formulation with an isocyanate index of 335 was as follows: LF polyol—50 pbw; n-pentane—13.7 pwb; blowing catalyst—3.3 pbw; and trimerization catalyst—1.0 pbw. The optimized rigid PIR foam formulation with an isocyanate index of 400 was as follows: LF polyol—75 pbw; n-pentane—15 pwb; blowing catalyst—3.5 pbw; and trimerization catalyst—1.5 pbw. The other reagents were selected according to Table 2.

The optimized rigid PIR foam formulations were evaluated against industrially produced rigid PIR foam materials from TENAX PANEL Ltd., prepared on a laboratory scale. TENAX PANEL Ltd. utilizes two distinct commercially available rigid PIR foam formulations, designated as A and B, to protect proprietary trade secrets. A summary of the key characteristics of the optimized rigid PIR foam is provided in Table 7.

The results demonstrate that the apparent density, thermal conductivity, and compression strength of the optimized rigid PIR foam are comparable to those of the commercially available formulations. These findings confirm that the developed PIR foam meets industry standards and is suitable for further upscaling in production. This alignment with existing industrial materials also suggests that the optimized formulations could seamlessly integrate into TENAX PANEL Ltd.’s production lines, ensuring compatibility and performance consistency.

## 4. Conclusions

This study successfully developed and optimized rigid PIR foam formulations, achieving high-performance thermal insulation materials suitable for industrial applications. By systematically analyzing key parameters such as the isocyanate index, blowing catalyst content, trimerization catalyst content, n-pentane content, and polyol composition, the optimized foams demonstrated a low thermal conductivity of 23.7 mW/(m·K), sufficient compression strength of 0.32 MPa, and well-controlled foaming kinetics. This study identified optimal foaming parameters, including a start time of 10.9 s, gel time of 37 s, and rise time of 57 s, ensuring compatibility with large-scale production.

The results highlight the critical influence of n-pentane and low-functionality polyol in reducing density and enhancing insulation properties, while catalysts played a pivotal role in fine-tuning foam expansion and reaction kinetics. Benchmarking against commercially available PIR foams confirmed that the optimized formulations align with current industry standards and manufacturing requirements. This research provides a strong foundation for the cost-effective production of high-quality PIR foams, contributing to advancements in thermal insulation materials through precise formulation control and material optimization.

While this study successfully optimized PIR foam formulations, certain limitations should be considered. This research was conducted under controlled laboratory conditions, and while the formulations were designed to align with industrial processes, additional validation under full-scale production settings is necessary. Moreover, this study primarily focused on thermal conductivity, compression strength, and foaming kinetics, but other important factors, such as long-term aging effects, dimensional stability, and fire resistance, require further investigation. Additionally, this study relied on conventional petrochemical-based polyols, and more research is needed to assess the feasibility of bio-based alternatives from an economic and performance standpoint.

## Figures and Tables

**Figure 1 materials-18-00881-f001:**
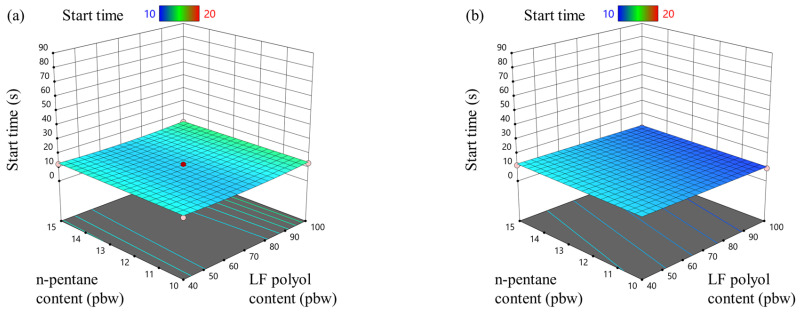
LF polyol and n-pentane influence on foaming start time for rigid PIR foam with an isocyanate index of (**a**) 335 and (**b**) 400; trimerization and blowing catalyst influence on foaming start time for rigid PIR foam with an isocyanate index of (**c**) 335 and (**d**) 400.

**Figure 2 materials-18-00881-f002:**
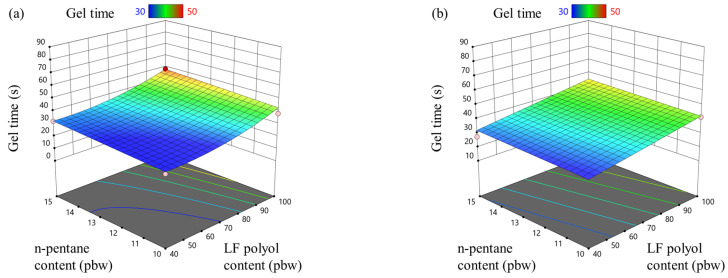
LF polyol and n-pentane influence on foaming gel time for rigid PIR foam with an isocyanate index of (**a**) 335 and (**b**) 400; trimerization and blowing catalyst influence on foaming gel time for rigid PIR foam with an isocyanate index of (**c**) 335 and (**d**) 400.

**Figure 3 materials-18-00881-f003:**
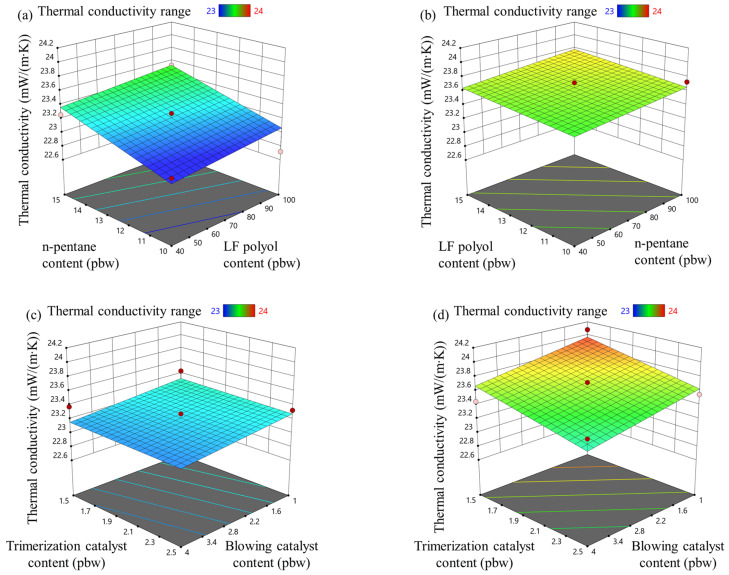
LF polyol and n-pentane influence on thermal conductivity for rigid PIR foam with an isocyanate index of (**a**) 335 and (**b**) 400; trimerization and blowing catalyst influence on thermal conductivity for rigid PIR foam with an isocyanate index of (**c**) 335 and (**d**) 400.

**Figure 4 materials-18-00881-f004:**
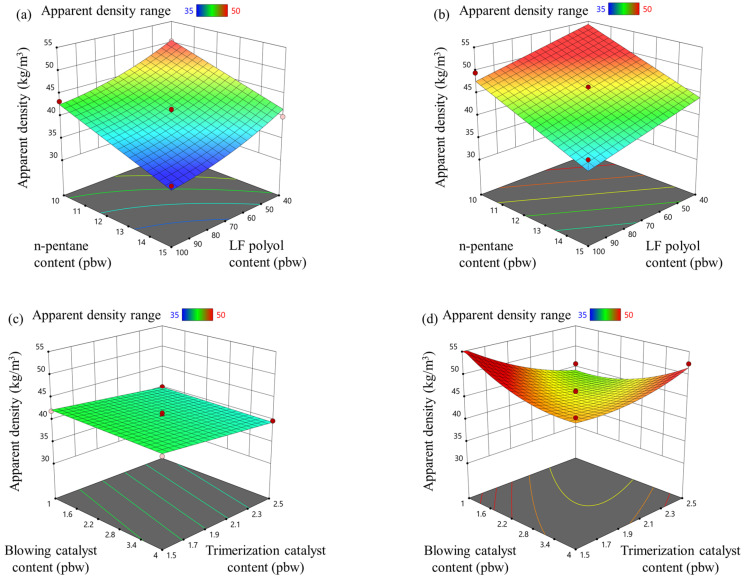
LF polyol and n-pentane influence on apparent density for rigid PIR foam with an isocyanate index of (**a**) 335 and (**b**) 400; trimerization and blowing catalyst influence on apparent density for rigid PIR foam with an isocyanate index of (**c**) 335 and (**d**) 400.

**Figure 5 materials-18-00881-f005:**
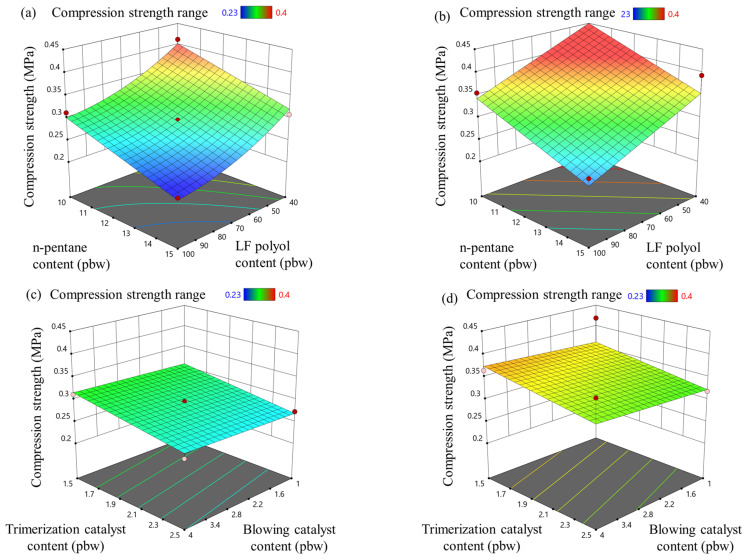
LF polyol and n-pentane influence on compression strength for rigid PIR foam with an isocyanate index of (**a**) 335 and (**b**) 400; trimerization and blowing catalyst influence on compression strength for rigid PIR foam with an isocyanate index of (**c**) 335 and (**d**) 400.

**Figure 6 materials-18-00881-f006:**
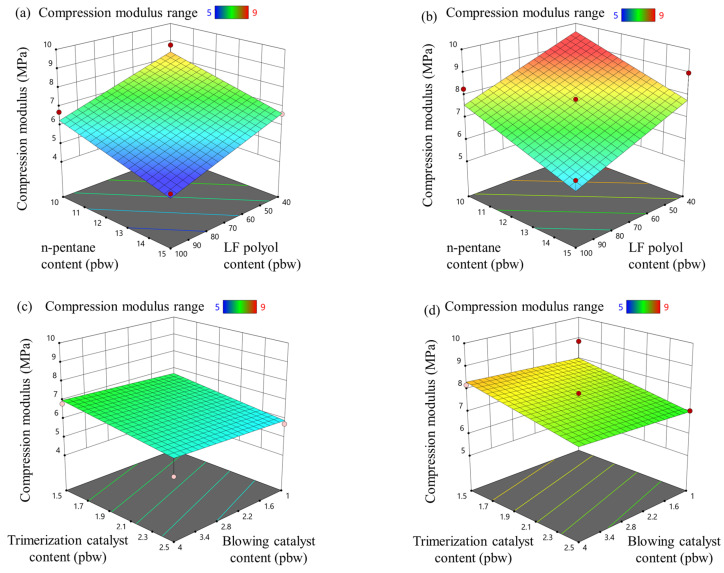
LF polyol and n-pentane influence on compression modulus for rigid PIR foam with an isocyanate index of (**a**) 335 and (**b**) 400; trimerization and blowing catalyst influence on compression modulus for rigid PIR foam with an isocyanate index of (**c**) 335 and (**d**) 400.

**Figure 7 materials-18-00881-f007:**
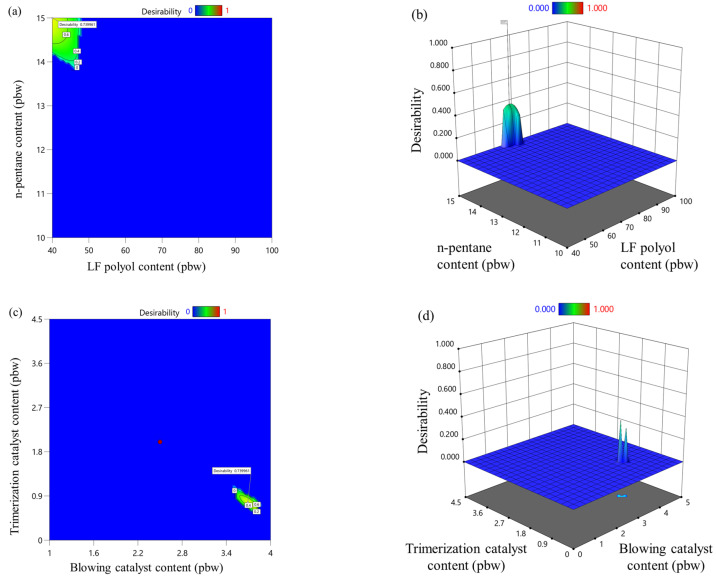
LF polyol and n-pentane influence on desirability for rigid PIR foam with an isocyanate index of (**a**) 335 and (**b**) 400; trimerization and blowing catalyst influence on desirability for rigid PIR foam with an isocyanate index of (**c**) 335 and (**d**) 400.

**Table 1 materials-18-00881-t001:** Comparison of thermal insulation properties of the most commonly used insulation materials.

Material Type	Thermal Conductivity, mW/(m·K)	Apparent Density, kg/m^3^	Ref.
Glass wool	30–40	125	[9,26]
Mineral wool	37–55	70–120	[9,27]
EPS	33–45	10–35	[20,28]
XPS	26–35	24–42	[22]
Open-cell rigid PU foam	34–63	10–35	[29,30,31,32]
Rigid PU foam	19–37	35–44	[33,34]
Rigid PIR foam	18–28	30–45	[22,35]
Cellulose	37–40	30–80	[22]
Cotton fibers	41.4	50	[34]
Hemp fiber	50	64	[24]

**Table 2 materials-18-00881-t002:** Rigid PIR foam formulations and changed parameters for RSM.

Components	PIR Foams with *II* = 335, pbw	PIR Foams with *II* = 400, pbw
Low functionality polyol	100–40	100–40
High functionality polyol	0–60	0–60
Flame retardant	11	11
Surfactant	3.0	3.0
Water	0.4	0.6
Blowing catalyst	1–4	1–4
Trimerization catalyst	1.5–2.5	1.5–2.5
Physical blowing agent	10–15	10–15
pMDI	230–255	275–304

**Table 3 materials-18-00881-t003:** The changed parameters, their identifier, and their lower and higher values.

Parameter	Identifier	Unit	Low	High	Coded Level		Mean
A	LF polyol content	pbw	40.00	100.00	−1 ↔ 40.00	+1 ↔ 100.00	70.00
B	Blowing catalyst	pbw	1.0000	4.00	−1 ↔ 1.00	+1 ↔ 4.00	2.50
C	Trimerization catalyst	pbw	1.50	2.50	−1 ↔ 1.50	+1 ↔ 2.50	2.00
D	n-pentane	pbw	10.00	15.00	−1 ↔ 10.00	+1 ↔ 15.00	12.50

**Table 4 materials-18-00881-t004:** Developed RSM models for rigid PIR foam characteristics and their ANOVA from foams with an isocyanate index 335.

Model Coefficients	Start Time, s	Gel Time, s	Rise Time, s	Tack Free Time, s	Temperature Peak in Foam, °C	Apparent Density, kg/m^3^	Thermal Conductivity, mW/(m·K)	Compression Strength, MPa	Compression Modulus, MPa
Intercept	+44.272	+97.577	+122.410	+101.447	+185.974	+127.691	+20.184	+0.7836	+14.5758
A	−0.199	−0.377	−0.304	+0.396	+0.050	−0.324	+0.030	−0.0045	−0.0350
B	−10.651	−16.478	−17.728	−16.556	−0.506	−0.183	−0.055	+0.0027	+0.0950
C	−7.167	−28.217	−35.350	−24.517	+3.533	−23.983	+0.914	−0.0388	−0.8033
D	+0.033	+1.067	+1.170	+0.457	−0.963	−5.273	+0.092	−0.0163	−0.3437
A·B				+0.048					
A·C				−0.127			−0.137		
A·D									
B·C	+1.933		+5.533	+3.633					
B·D		+4.833							
C·D						+1.740			
A2	+0.0019	+0.0046	+0.0041		−0.0010	+0.0014		+0.00002	
B2	+0.807			+0.398					
C2									
D2									
ANOVA									
F-value	48.01	44.13	75.80	57.04	60.69	47.84	5.65	40.67	30.34
*p*-value	<0.0001	<0.0001	<0.0001	<0.0001	<0.0001	<0.0001	<0.0027	<0.0001	<0.0001
R^2^	0.949	0.933	0.960	0.964	0.938	0.938	0.611	0.910	0.859

A—LF-polyol content; B—blowing catalyst content; C—trimerization catalyst content; D—physical blowing agent content.

**Table 5 materials-18-00881-t005:** Developed RSM models for rigid PIR foam characteristics and their ANOVA from foams with an isocyanate index 400.

Model Coefficients	Start Time, s	Gel Time, s	Rise Time, s	Tack Free Time, s	Temperature Peak in Foam, °C	Apparent Density, kg/m^3^	Thermal Conductivity, mW/(m·K)	Compression Strength, MPa	Compression Modulus, MPa
Intercept	+41.028	+70.900	+85.216	+97.127	+193.591	+181.555	+24.099	+0.779	+16.450
A	−0.028	+0.459	+0.196	+0.165	−0.061	−0.111	+0.002	−0.0018	−0.034
B	−11.968	−9.351	−6.302	−4.467	−2.507	−15.502	−0.099	+0.0044	+0.097
C	−7.700	−28.850	−19.450	−21.503	+0.794	−60.328	−0.338	−0.038	−1.024
D	+0.080	+0.585	+0.429	+0.057	−0.749	−5.034	+0.025	−0.019	−0.367
A·B		−0.096							
A·C									
A·D									
B·C	+1.989	+4.950			+0.844	+5.077			
B·D									
C·D						+1.492			
A2									
B2	1.085					+1.048			
C2						+6.296			
D2									
ANOVA									
F-value	33.54	25.77	30.19	17.05	22.65	27.19	6.42	16.06	13.08
*p*-value	<0.0001	<0.0001	<0.0001	<0.0001	<0.0001	<0.0001	<0.0027	<0.0001	<0.0001
R^2^	0.923	0.901	0.864	0.782	0.863	0.940	0.588	0.772	0.744

A—LF-polyol content; B—blowing catalyst content; C—trimerization catalyst content; D—physical blowing agent content.

**Table 6 materials-18-00881-t006:** Optimization parameters for rigid PIR foams with an isocyanate index of 335 and 400.

Parameter Name	Goal	Lower Limit	Upper Limit	Impotence
A: Polyol content	is in range	40	100	3
B: Blowing catalyst	is in range	0	5	3
C: Trimerization catalyst	is target = 2	0	4.5	3
D: n-pentane	is in range	10	15	3
Start time	is target = 10.9	10.4	11.4	3
Gel time	is target = 37	35	39	3
Rise time	is target = 57	55	59	3
Tack free time	none	38.6	81.4	3
Temperature peak	none	172.7	182.7	3
Apparent density	minimize	38	42	5
Thermal conductivity	minimize	22.72	23.55	5
Compression strength	maximize	0.224	0.412	3
Compression modulus	maximize	4.77	8.76	3

**Table 7 materials-18-00881-t007:** Common characteristics of optimized rigid PIR foam with isocyanate indices of 335 and 400 and the comparison to commercially available rigid PIR foam at TENAX PANEL Ltd.

Formulation	A	B	TPIR335_O2	TPIR400_O2
Apparent density, kg/m^3^	41.3	37.6	43.3	41.6
Thermal conductivity, mW/(m·K)	23.13	24.02	23.7	24.11
σ_z_, MPa	0.320 ± 0.021	0.269 ± 0.010	0.323 ± 0.009	0.294 ± 0.006
E_compZ_, MPa	7.21 ± 0.79	5.94 ± 0.20	7.20 ± 0.23	6.86 ± 0.13
σ_x_, MPa	0.210 ± 0.013	0.196 ± 0.013	0.223 ± 0.008	0.228 ± 0.009
E_compX_, MPa	4.23 ± 0.36	3.93 ± 0.57	3.91 ± 0.17	3.75 ± 0.21

z—denotes testing parallel to the foaming direction; x—denotes testing perpendicular to the foaming direction.

## Data Availability

The authors state that the presented data will be available on request by email.

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
