# Peer review of "Modeling Key Characteristics of Rigid Polyisocyanurate Foams to Improve Sandwich Panel Production Process"

_materials, 2025, doi:10.3390/ma18040881_

Round 1
Reviewer 1 Report
Comments and Suggestions for Authors
The manuscript is interesting and focusing on the thermal insulation foams.
The introduction contains lumped references. Please access every reference with at least a half sentence.
The manuscript is interesting and the choosen method should be acceptable, however some key features are missing.
Please add table containing the factors, and their identifier, lower, and higher values. Extend it with the scaled values.
How many parallel experiments were performed.
The detailed measurement, and specimen table should be added as well, as the comparison of the measurement, and the calculated values in a graph.
Table 5 is not clear, what impotence means in this table?
It is not clear how the optimization was performed, which program was used? How the authors ensured they found a global optimum?
The key findings should be described in detail in the conclusion as well.
Author Response
Answers to Reviewer 1
- The manuscript is interesting and focusing on the thermal insulation foams.
Thank you kindly.
- The introduction contains lumped references. Please access every reference with at least a half sentence.
The references have been improved accordingly.
- The manuscript is interesting and the choosen method should be acceptable, however some key features are missing. Please add table containing the factors, and their identifier, lower, and higher values. Extend it with the scaled values.
The manuscript has been improved accordingly.
The changed parameters, their identifier, their lower and higher values and coded levels are summarized in Table 3.
|
Parameter |
Identifier |
unit |
Low |
High |
Coded level |
Mean |
|
|
A |
LF polyol content |
pbw |
40.00 |
100.00 |
-1 ↔ 40.00 |
+1 ↔ 100.00 |
70.00 |
|
B |
Blowing catalyst |
pbw |
1.0000 |
4.00 |
-1 ↔ 1.00 |
+1 ↔ 4.00 |
2.50 |
|
C |
Trimerization catalyst |
pbw |
1.50 |
2.50 |
-1 ↔ 1.50 |
+1 ↔ 2.50 |
2.00 |
|
D |
n-pentane |
pbw |
10.00 |
15.00 |
-1 ↔ 10.00 |
+1 ↔ 15.00 |
12.50 |
Table 3. The changed parameters, their identifier and their lower and higher values
- How many parallel experiments were performed.
At least three parallel measurements were done for the FOAMAT testing. The data points used for RSM was taken as an average from three measurements. Compression strength was tested for six parallel samples but thermal conductivity was measured for two samples due to large sample size. The apparent density of the PIR foams was measured for each compression strength sample, i.e. 12 samples per series.
- The detailed measurement, and specimen table should be added as well, as the comparison of the measurement, and the calculated values in a graph.
Thank you for the suggestion. The comparison between the calculated value and the measured value can be seen as the red/pink dots in the RSM graphs. The dost are an average value between at least three parallel samples (except thermal conductivity data).
The supplementary data has been improved by adding Table S1 and Table S2 which summarizes the changed factor values and all of the determined ten measured responses of the RSM.
- Table 5 is not clear, what impotence means in this table?
The importance parameter for the RSM optimization has been explained in the text of the manuscript:
A different value of importance was assigned for each factor or response to prioritize the more essential ones. The importance was scaled from 1 to 5, with 1 being the least significant and 5 being the most important. The overall function becomes 0 if the response or factors fall outside their desirability range.
- It is not clear how the optimization was performed, which program was used? How the authors ensured they found a global optimum?
The optimization was carried out using Design Expert software, which features a built-in function for optimizing RSM models. The objective of the study was not to achieve a global optimum but rather to find an optimal solution within the constraints defined by the experimental design, as outlined in Tables 2 and 3. The focus was on optimizing the rigid PIR foam formulation while ensuring the constraints were feasible for industrial production.
Several practical considerations guided the optimization process. The amount of catalyst was carefully limited to maintain the cost-effectiveness of the material. Additionally, the apparent density of the PIR foam, which is directly proportional to compression strength and inversely proportional to thermal conductivity, was restricted to a maximum of 40 kg/m³ to avoid compromising economic viability.
Finally, the foaming kinetic parameters were adjusted to fit the requirements of the production line. These parameters needed to align with the line speed and the thickness of the PIR sandwich panels being manufactured. This approach ensured that the optimization was tailored to real-world industrial applications, rather than pursuing a theoretical global optimum.
- The key findings should be described in detail in the conclusion as well.
The conclusions have been improved accordingly.
This study successfully developed and optimized rigid PIR foam formulations, achieving high-performance thermal insulation materials suitable for industrial applications. By systematically analyzing key parameters such as the isocyanate index, blowing catalyst content, trimerization catalyst content, n-pentane content, and polyol composition, the optimized foams demonstrated a low thermal conductivity of 23.7 mW/(m·K), sufficient compression strength of 0.32 MPa, and well-controlled foaming kinetics. The study identified optimal foaming parameters, including a start time of 10.9 seconds, gel time of 37 seconds, and rise time of 57 seconds, ensuring compatibility with large-scale production.
The results highlight the critical influence of n-pentane and low-functionality polyol in reducing density and enhancing insulation properties, while catalysts played a pivotal role in fine-tuning foam expansion and reaction kinetics. Benchmarking against commercially available PIR foams confirmed that the optimized formulations align with current industry standards and manufacturing requirements. This research provides a strong foundation for the cost-effective production of high-quality PIR foams, contributing to advancements in thermal insulation materials through precise formulation control and material optimization.
While this study successfully optimized PIR foam formulations, certain limitations should be considered. The research was conducted under controlled laboratory conditions, and while the formulations were designed to align with industrial processes, additional validation under full-scale production settings is necessary. Moreover, the study primarily focused on thermal conductivity, compression strength, and foaming kinetics, but other important factors, such as long-term aging effects, dimensional stability, and fire resistance, require further investigation. Additionally, the study relied on conventional petrochemical-based polyols, and more research is needed to assess the feasibility of bio-based alternatives from an economic and performance standpoint.
Reviewer 2 Report
Comments and Suggestions for Authors
I suggest to use superscript unit format to avoid brackets (e.g. abstract).
Regarding the drawbacks of foam insulations, you should maybe mention that most of these products are fossil-fuel based / fossil ressources based materials, and thus have a questionable environmental footprint. This is - imho - an even stronger drawback than the flammability.
Can you maybe go to the unit W/(mK) instead of mW/(mK), which is uncommon in the AEC-domain.
Is it necessary to mention the companies producing your test facilities including their (R) sign? This is a scientific paper, not a advertisment. Name the devices, reference them, but do it in a decent form.
Figure 1: Too small to read (text font size), additionally, can you please avoid 3D graphs. These do not provide any benefit for readers at all - you can not simply take out values from it.; THis comment is also true for other figures in the paper.
The conclusion section of your paper is comparably short. Please add a section "Future Research" and a section "Limitations of this study" to the conclusions. Moreover - please get rid of these commercial product expression (TENAX PANEL Ltd.) - you want to write a scientific paper and not perform lobbying for a product.
Author Response
Answers to Reviewer 2
- I suggest to use superscript unit format to avoid brackets (e.g. abstract).
Thank you for your suggestion. However, we have chosen to retain the current unit format with brackets (e.g., mW/(m·K)) for consistency with common practices in similar studies and ease of readability. We believe this format is clear and widely recognized within the field.
- Regarding the drawbacks of foam insulations, you should maybe mention that most of these products are fossil-fuel based / fossil ressources based materials, and thus have a questionable environmental footprint. This is - imho - an even stronger drawback than the flammability.
Thank you for the suggestion. The introduction of the manuscript has been improved accordingly: “Although there are plenty of studies describing rigid PIR foam development from bio-based feedstocks such as rapeseed oil [16], tall oil fatty acids [17], date seed oil [18] their wider application on an industrial scale is limited due to their cost.”
We agree that using non-fossil-based materials would be beneficial, as it could help reduce the carbon footprint of PIR foams. However, under current market conditions, none of the bio-based polyols available at a large scale (50–100 tonnes per month) fall within an economically feasible price range. To develop a cost-effective rigid PIR foam polyol component, the price of neat bio-based polyol should be between 1.5–1.8 EUR/kg. However, TENAX PANEL Ltd. has received offers for rapeseed oil-based polyols at 2.5–3.0 EUR/kg and cashew nut-based polyols at 3.0–3.5 EUR/kg. At present, we do not see how customers would be willing to absorb such a price difference, particularly in the absence of regulatory pressure to replace petrochemical feedstocks.
- Can you maybe go to the unit W/(mK) instead of mW/(mK), which is uncommon in the AEC-domain.
Thank you for your suggestion. However, we have chosen to retain the unit mW/(m·K) as it is commonly used in the field of thermal insulation materials. This format ensures consistency with existing literature and facilitates direct comparison with similar studies. We appreciate your understanding.
- Is it necessary to mention the companies producing your test facilities including their (R) sign? This is a scientific paper, not a advertisment. Name the devices, reference them, but do it in a decent form.
Thank you for your comment. The FOAMAT® device is a trademarked instrument, and the other names mentioned refer to specific instrument models. We have chosen to include these names as it aligns with the requirements set by the journal editor in our previous publications. This ensures consistency and clarity in reporting the methodology while maintaining scientific accuracy.
- Figure 1: Too small to read (text font size), additionally, can you please avoid 3D graphs. These do not provide any benefit for readers at all - you can not simply take out values from it.; THis comment is also true for other figures in the paper.
We believe that 3D graphs allow for easier comprehension of data that is shown in three dimensions. Furthermore, the experiment is relatively complicated where four different factors influence two different rigid PIR formulation characteristics. Thus, we chose to use 3D graphs. We agree that it is difficult to take data from the plots. It would be even more difficult to read specific data from a topological-type graph, nevertheless, it is shown as coloured lines underneath the response surface plot. To read the measured data we have added Table S1 and Table S2 to the supplementary information which contains all the changed parameters and average values of the measured responses. Lastly, we have changed the figures in the articly with an increased font size.
- The conclusion section of your paper is comparably short. Please add a section "Future Research" and a section "Limitations of this study" to the conclusions. Moreover - please get rid of these commercial product expression (TENAX PANEL Ltd.) - you want to write a scientific paper and not perform lobbying for a product.
Thank you for the suggestion the conclusions have been revised to include the limitations of the study and plans for further research on the flammability study of the developed PIR foam formulation and possible increase of the sustainability by using bio-based polyols.
Revised conclusions:
This study successfully developed and optimized rigid PIR foam formulations, achieving high-performance thermal insulation materials suitable for industrial applications. By systematically analyzing key parameters such as the isocyanate index, blowing catalyst content, trimerization catalyst content, n-pentane content, and polyol composition, the optimized foams demonstrated a low thermal conductivity of 23.7 mW/(m·K), sufficient compression strength of 0.32 MPa, and well-controlled foaming kinetics. The study identified optimal foaming parameters, including a start time of 10.9 seconds, gel time of 37 seconds, and rise time of 57 seconds, ensuring compatibility with large-scale production.
The results highlight the critical influence of n-pentane and low-functionality polyol in reducing density and enhancing insulation properties, while catalysts played a pivotal role in fine-tuning foam expansion and reaction kinetics. Benchmarking against commercially available PIR foams confirmed that the optimized formulations align with current industry standards and manufacturing requirements. This research provides a strong foundation for the cost-effective production of high-quality PIR foams, contributing to advancements in thermal insulation materials through precise formulation control and material optimization.
While this study successfully optimized PIR foam formulations, certain limitations should be considered. The research was conducted under controlled laboratory conditions, and while the formulations were designed to align with industrial processes, additional validation under full-scale production settings is necessary. Moreover, the study primarily focused on thermal conductivity, compression strength, and foaming kinetics, but other important factors, such as long-term aging effects, dimensional stability, and fire resistance, require further investigation. Additionally, the study relied on conventional petrochemical-based polyols, and more research is needed to assess the feasibility of bio-based alternatives from an economic and performance standpoint.
Reviewer 3 Report
Comments and Suggestions for Authors
The submitted article is dealing with the study of polyisocyanurate foam for insulation. The introduction is well organized and lists usual insulation materials with their pros and cons. The methodology that was used is appropriate, and supported by proper statistical analysis. The results are well discussed and connected not only to laboratory scale experiments, but to industry also. The use of language is proper; the manuscript is well written and organized, and very easy to follow.
I have very few points that are stated in the following, and all of them are considered as minor.
1. Line 120. A “a” appears in the last sentence.
2. Line 141. Please, add a space between values and units.
3. Enhance the manuscript with more references. It is noted that after line 78, referencing is scarce, not only in the introduction, but also in results and discussion; in the latter, less than 10 references were used.
4. Figures. Please, use a larger font size for labeling your axes and annotations, it is quite difficult to read them in a A4 print.
5. There is supposed to be a supplementary file, but nothing was found through the files I had access to.
Author Response
Answers to Reviewer 3
The submitted article is dealing with the study of polyisocyanurate foam for insulation. The introduction is well organized and lists usual insulation materials with their pros and cons. The methodology that was used is appropriate, and supported by proper statistical analysis. The results are well discussed and connected not only to laboratory scale experiments, but to industry also. The use of language is proper; the manuscript is well written and organized, and very easy to follow.
Thank you kindly for the review.
I have very few points that are stated in the following, and all of them are considered as minor.
- Line 120. A “a” appears in the last sentence.
We have corrected the text line.
- Line 141. Please, add a space between values and units.
Thank you for the keen notice, the space has been added.
- Enhance the manuscript with more references. It is noted that after line 78, referencing is scarce, not only in the introduction, but also in results and discussion; in the latter, less than 10 references were used.
Thank you for the comment, we have improved the manuscript accordingly.
- Please, use a larger font size for labeling your axes and annotations, it is quite difficult to read them in a A4 print.
The fonts of the figures have been increased. Moreover, the data points used in the figures have been added to supplementary information in Table 1 and Table 2.
- There is supposed to be a supplementary file, but nothing was found through the files I had access to.
Yes, the supplementary information file was added to the journal system.
Round 2
Reviewer 1 Report
Comments and Suggestions for Authors
The authors addressed all my comments, the manuscript can be accepted.